# Experimental Study on Near-Wall Laser-Induced Cavitation Bubble Micro-Dimple Formation on 7050 Aluminum Alloy

**Yupeng Cao** [1,2]**, Ranran Hu** [1]**, Weidong Shi** [1,]*** and Rui Zhou** [3,]***

[1] School of Mechanical Engineering, Nantong University, Nantong 226019, China; cyp19812004@ntu.edu.cn (Y.C.); 2109310001@stmail.ntu.edu.cn (R.H.)

[2] 3D Printing Technology Research Institute, Nantong Institute of Technology, Nantong 226001, China

[3] School of Electronic Information, Nantong University, Nantong 226019, China

[*] Correspondence: wdshi@ntu.edu.cn (W.S.); 2110510005@stmail.ntu.edu.cn (R.Z.)

**Abstract:** To investigate the feasibility and formation laws of fabricating micro-dimples induced by near-wall laser-induced cavitation bubble (LICB) on 7050 aluminum alloy. A high-speed camera and a fiber-optic hydrophone system were used to capture pulsation evolution images and acoustic signals of LICB. Meanwhile, a three-dimensional profilometer was employed to examine the contour morphology of the surface micro-dimple on the specimen. The results show that at an energy level of 500 mJ, the total pulsation period for the empty bubble is 795 μs, with individual pulsation periods of 412.5 μs, 217 μs, and 165 μs for the first, second, and third cycles, respectively, with most energy of the laser and bubble being consumed during the first evolution period. Under the synergy of the plasma shock wave and collapse shock wave, a spherical dimple with a diameter of 450 μm is formed on the sample surface with copper foil as the absorption layer. A model of micro-dimple formed by LICB impact is established. As the energy increases, the depth of the surface micro-dimple peaks at an energy of 400 mJ and then decreases. The depth of the surface micro-dimple increases with the increase in the number of impacts; the optimal technology parameters for the micro-dimple formation by LICB impact are as follows: the absorption layer is copper foil, the energy is 400 mJ, and the number of impacts is three.

**Keywords:** laser-induced cavitation bubble (LICB); absorption layer; surface morphology



## 1. Introduction

Surface texturing (ST) is a surface modification technique that effectively reduces frictional wear and enhances the performance of contact surfaces [1]. Traditional surface texturing techniques such as electrochemical discharge machining [2,3], reactive ion etching [4], and photolithography [5] have not fully met the increasing demands for the variety, scale, and precision of surface microstructures. In recent years, laser processing technology, characterized by its simplicity, controllability, high precision, and efficiency, has become an important research tool. Among its applications, laser surface texturing (LST) technology leverages the thermal effects of lasers, which induce micro-vaporization and ablation on the material surface, occasionally resulting in the formation of micro-cracks [6]. Laser peening texturing (LPT) technology utilizes the mechanical effects of laser shock waves to create micro-dimple arrays on the material surface, effectively mitigating the adverse effects of laser thermal effects, which can cause the accumulation of molten material at high temperatures around the edges of structures. However, this technology still has drawbacks, including large dimple sizes and high energy dissipation [7]. The application of cavitation water jet peening (CWJP) involves generating cavitation bubbles through the shearing effect of a high-speed water jet in an artificially submerged environment, leading to material removal [8–10]. Research conducted by Han et al. [11] suggested that due to the inclination error in vertical jetting, uneven removal depths were observed in the pits generated during the immersion-based pulsed cavitation jet peening process.

Laser-induced cavitation bubble (LICB) represents a new technology developed from the laser ablation technology underwater. The principle underlying LICB involves the generation of plasma through laser breakdown of the liquid threshold. Subsequently, the plasma continuously absorbs laser energy to expand into a bubble. Due to the imbalance in the internal and external pressure, the bubble undergoes periods of expansion, contraction, and eventual collapses [12–14]. When the bubble collapses near the wall surface, it generates collapse shock wave and micro-jet [15–18]. Long et al. [19,20] investigated the influence of different liquid media and depth on the evolution of LICB and surface ablation. In order to prevent ablation, Nie et al. [21] altered the direction of laser irradiation to be parallel to the sample, significantly reducing the impact force induced on the sample. To explore the strengthening effect of LICB on material, Gu et al. [22] demonstrated that LICB increased the residual stress and microhardness on the material surface; Zhang et al. [23] indicated that LICB could reduce the surface roughness of the material. LICB has been shown to modify the surface of the material, concurrently fabricating micro-dimples through surface deformation. Hanke and Gonzalez [24,25] investigated the surface damage and mass loss of steel and brass caused by a single LICB. Ren et al. [26] studied the phase and microstructure changes in an aluminum alloy material treated by laser cavitation peening. Ye et al. [27] fabricated randomly distributed micro-dimples on the surface of 304 L stainless steel through underwater laser ablation texturing and investigated the effects of laser parameters on micro-dimple formation. At present, the main research conducted by related scholars in this field is focused on the material surface modification of LICB, while little attention is paid to the irregular micro-dimple induced. Research on the fabrication of micron grade dimple by LICB has not been reported yet.

This study carried out a single-point impact test on the surface of 7050 aluminum alloy by LICB technology. A high-speed camera was used to capture the entire life cycle of the cavitation bubble near the wall surface, and the feasibility of fabricating a micro-dimple by LICB was verified. The surface morphology of the dimple was observed by a three-dimensional profilometer. The relationship between different technology parameters (absorption layer, laser energy, number of impacts) and the geometric characteristics of the micro-dimple was explored. The above research content enables the quantitative loading and precise control of the micro-texturing formation process and provides a new technological approach for laser micro-texturing.

## 2. Experimental Setup and Methods

### 2.1. Experimental Setup

The experimental setup for LICB fabricating micro-dimple formation is shown in Figure 1. This platform comprised a Nd: YAG solid-state pulsed laser (Nimma Extra, Beamtech, Shenzhen, China), a high-speed camera (V2012, Phantom, Los Angeles, CA, USA), a fiber-optic hydrophone system (V2, PA, Santa Clara, CA, USA), a high-performance oscilloscope (DSO91304A, Agilent, Santa Clara, CA, USA), optical components, and a workbench.

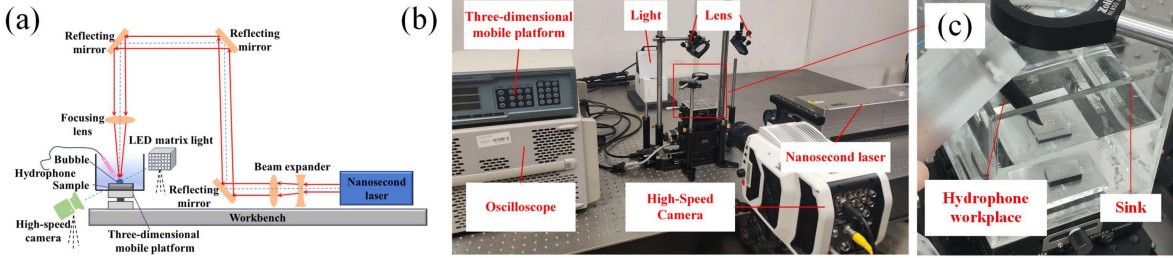

**Figure 1.** Platform schematic diagram and field diagram of LICB. (**a**) Platform diagram; (**b**) device site diagram; (**c**) hydrophone workplace.

## 2.2. Sample and Specimen Pre-Treatment

The 7050-T7451 aluminum alloy was selected as the specimen due to its excellent formability and corrosion resistance. The primary chemical composition of the alloy is provided in Table 1. The sample was processed into specimens measuring 20 mm × 20 mm × 5 mm through wire cutting. Subsequently, the specimens underwent progressive polishing, ranging 60#~2500# sandpapers, followed by polishing with a 0.5 μm diamond polishing paste until the achievement of a mirror-like surface. Finally, the specimens were cleaned and dried in an ultrasonic cleaner.

**Table 1.** Chemical composition of 7050-T7451 aluminum alloys (mass fraction, %).

| Element | Si | Fe | Cu | Mn | Mg | Zn |
|---------|------|------|---------|-----|---------|---------|
| Value | 0.12 | 0.15 | 2.0–2.6 | 0.1 | 1.9–2.6 | 5.6–6.7 |

## 2.3. Experimental Process and Parameters

It can be observed from Figure 1 that the laser emitted from the laser device underwent a fourfold expansion, which was then elevated through a beam-raising frame to raise the optical path. Subsequently, it was vertically focused by a convex lens into a high-transmittance acrylic transparent water sink. The sink was placed on an electric three-dimensional moving platform (CZF 20–120, Zolix, Beijing, China), with 500 mL of deionized water inside. By adjusting the three-dimensional moving platform, the laser was focused on the specimen surface, resulting in a focal spot diameter of 0.5 mm. Using a high-speed camera with a resolution of 256 × 256 pixels, LICB evolution was recorded at a speed of 400,000 frames per second. A synchronization signal generator triggered the high-speed camera to start capturing images when the laser emitted pulses, setting the start of the bright white light emitted by laser breakdown in water at 0 μs. An LED matrix light source (OS200, Orymate, Shenzhen, China) provided background illumination for the recordings. The hydrophone was positioned 10 cm away from the laser interaction point and oriented at 45° to the specimen surface. The acoustic signals captured by the hydrophone were converted into voltage signals by a high-performance oscilloscope, with the voltage signals representing the shock wave signals induced during the evolution of LICB. The specimen underwent LICB impact to generate micro-dimples on its surface. The three-dimensional morphology and two-dimensional profiles of these micro-dimples were examined by a non-contact optical profilometer (Usurf, NanoFocus, Oberhausen, Germany).

To investigate the technology of LICB fabricate micro-dimple formation, experiments were conducted with different absorption layers and laser technology parameters. Specifically, absorption layers of 100 μm black paint, 100 μm copper foil, and no absorption layer were used. The specific parameters for LICB impact are detailed in Table 2, and the paths of single dimple at different energy levels are shown in Figure 2.

**Table 2.** Parameters of LICB impact micro-dimple test.

| Test Parameter | Value |
|----------------|-------|
| Laser wavelength/nm | 1064 |
| Laser pulse width/ns | 9 |
| Laser frequency/Hz | 5 |
| Spot diameter/mm | 0.5 |
| Laser energy/mJ | 100, 200, 300, 400, 500 |
| Shock times | 1, 2, 3 |
| Temperature | 25 °C |
| Air humidity | 50% |

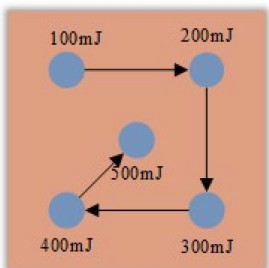

**Figure 2.** Schematic diagram of paths with different laser energy impact.

### 3. Result and Discussion

#### 3.1. Evolution of Cavitation Bubble

The evolution of a cavitation bubble with 500 mJ laser energy is shown in Figure 3. The laser power density exceeding the breakdown threshold of water induced the generation of high-temperature and high-pressure plasma, which rapidly expands near the sample, forming a hemispherical bubble [28–30]. It can be observed from Figure 3 that laser irradiation on the sample surface resulted in the generation of plasma, which rapidly expanded, leading to the formation of an initial cavitation bubble. Due to the pressure difference inside and outside the bubble, the bubble began to undergo evolution. During the initial evolution period, the bubble expanded to its maximum radius (see 2.5 μs–202.5 μs), then contracted until it finally collapsed (see 205 μs–412.5 μs). During the contraction phase, different velocities on the upper and lower surfaces of the bubble, due to the solid boundary, resulted in collapse when the upper surface reached the lower surface. The shock wave generated by the collapse of the bubble acted on the surface of the 7050 aluminum alloy, with the period of the first bubble evolution being 412.5 μs. At the end of the first evolution period, the collapse shock wave led to the formation of an irregular hemispherical bubble [31,32]. During the second evolution period, the bubble irregularly expanded and contracted, ranging from expansion (see 415 μs–542.5 μs) to contraction–collapse (see 545 μs–630 μs), with a shortened evolution period duration of 217.5μs. The third evolution period occurred from expansion (see 632.5 μs–715 μs) to contraction (see 717.5 μs–795 μs), with an evolution period duration of 165 μs. The LICB underwent three periodic evolutions before eventually dissipating in the liquid.

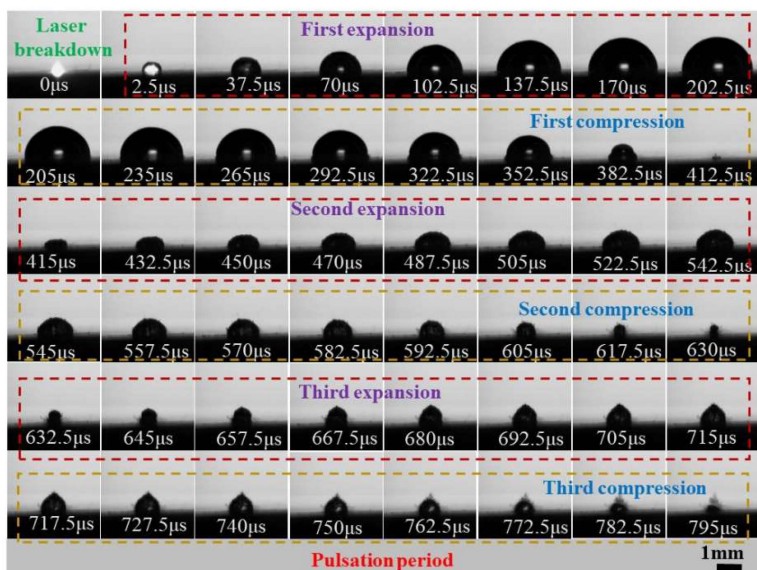

**Figure 3.** Evolution of bubble shapes with 500 mJ laser energy.

The acoustic signal of the cavitation bubble with 500 mJ laser energy is shown in Figure 4. It is observed from Figure 4 that the acoustic signal exhibits two peaks, representing the laser-induced plasma shock wave and the bubble collapse shock wave, respectively. Due to the influence of the near wall surface, the bubble collapse shock wave was attenuated [33], while the plasma shock wave generated by photoionization was stronger than the collapse shock wave of the cavitation bubble. In addition to these two prominent peaks, there were also some minor peaks in the acoustic signal. According to the studies by Lu et al. [34], it can be inferred that these minor peaks were caused by weak shock waves generated during the evolution of the cavitation bubble and background noise signals from the hydrophone. These minor peaks were considered unavoidable interference signals and can be safely disregarded.

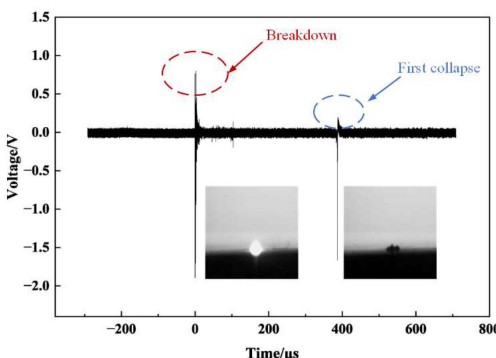

**Figure 4.** Diagram of acoustic signal with 500 mJ energy.

The relationship between the evolution size and time of cavitation bubble with 500 mJ laser energy is shown in Figure 5. Figure 5a presents the dynamic radius of the cavitation bubble, while Figure 5b shows the maximum radius of the cavitation bubble at different periods along with the evolution time. It can be observed from Figure 5 that the total duration of the three evolution periods of the cavitation bubble is 795 μs, with the durations of the first, second, and third evolution periods being 412.5 μs, 217.5 μs, and 165 μs, respectively. The decrement rates are 47.27% and 24.14%, with the rate of decrease gradually slowing down. The maximum radius of the cavitation bubble during the first, second, and third evolution periods are 3.03 mm, 1.60 mm, and 1.39 mm, respectively. The reduction rates were 47.19% and 13.13%, with the rate of decline gradually slowing down. The process of LICB underwent three periodic evolutions, with each successive evolution decreasing in duration and maximum radius until the cavitation bubble collapsed. It can be inferred that most of the laser energy and bubble energy were consumed during the first evolution period, and the impact of the shock wave generated during the third collapse was relatively small.

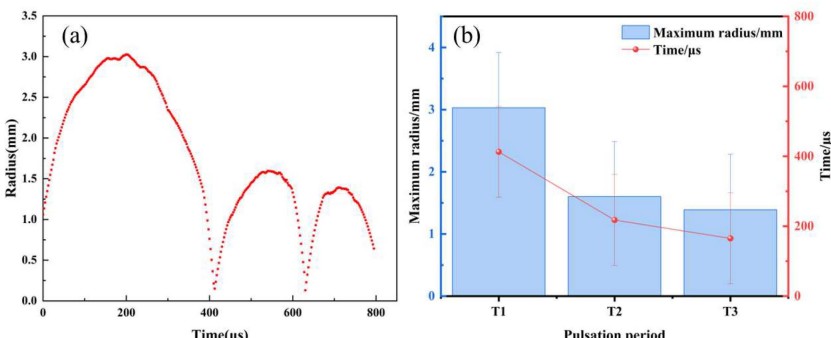

**Figure 5.** The relationship between the evolution size and time of cavitation bubble with 500 mJ laser energy. (**a**) Dynamic radius; (**b**) maximum dimple radius and evolution time of each period. T1: the first evolution period; T2: the second evolution period; T3: the third evolution period.

### 3.2. Modeling of LICB Impact on Micro-Dimple Formation

The surface morphology and the central cross-sectional profile of LICB micro-dimples are shown in Figure 6. High-power laser irradiation onto the sample surface rapidly induces plasma formation and expansion outward at an ultra-high speed. Under the constraint of the water layer on the sample surface, a high-pressure shock wave was generated and propagated into the sample. The force exerted by the laser plasma shock wave acted on the sample, causing plastic deformation under high strain rates. The expanded plasma evolved into a cavitation bubble. Through multiple expansion–contraction–collapse periods, the shock wave generated after collapse again acted on the sample, resulting in the formation of a dimple on the sample surface due to the synergy of the plasma shock wave and the collapse shock wave, as shown in Figure 6a. The laser energy and the shock wave generated by the cavitation bubble were propagated in the form of a spherical wave [35]. The bottom of the dimple formed on the sample surface by LICB impact was relatively sharp, as shown in Figure 6b. Therefore, it can be inferred that the shock wave induced by LICB exhibited a Gaussian distribution in spatial dimensions.

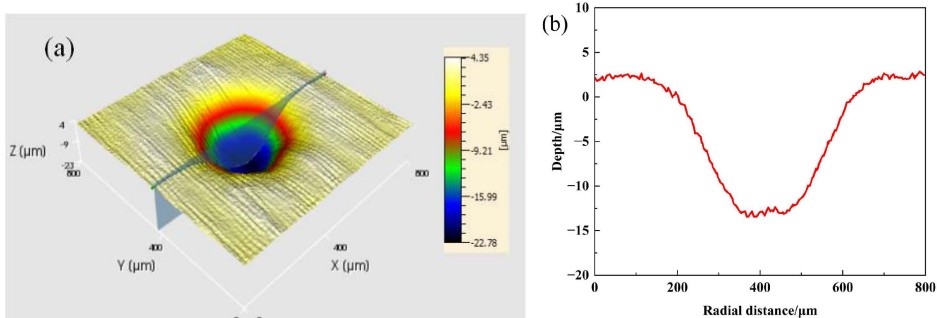

**Figure 6.** Surface morphology of micro-dimple with LICB. (**a**) Three-dimensional morphology; (**b**) central cross-sectional profile.

The schematic diagram showing the principle of LICB impact on micro-dimple formation is shown in Figure 7. It can be seen from Figure 3 that the laser-induced bubble underwent three periodic evolutions from initial formation to final collapse and dissipation, constituting three stages of micro-dimple formation. High-powered laser focused on piercing the local liquid in the medium, leading to rapid plasma formation on the sample surface and outward expansion (see 0 μs in Figure 3). Constrained by the surface water layer, a high-pressure shock wave was generated and propagated into the sample, inducing plastic deformation on the sample surface, as shown in Figure 7a. Under the constraint of the surface water layer on the sample, the plasma evolved into a hemispherical bubble (see 2.5 μs–202.5 μs in Figure 3). At this stage, due to the plasma inside the bubble, its pressure exceeded that of the surrounding water medium, resulting in a pressure differential. Subsequently, as the bubble expanded to its maximum radius, its internal pressure dropped below that of the water medium, leading to the initiation of shrinkage and eventual collapse of the bubble (see 205 μs–412.5 μs in Figure 3), as shown in Figure 7b. It can be seen from Figure 6 that the formation of the surface dimple on the specimen is under the combined influence of the plasma shock wave and the collapse shock wave of the bubble. The bubble underwent multiple periods of expansion and contraction, and the shock wave generated upon its collapse caused localized plastic deformation on the specimen surface, resulting in a micro-dimple, as shown in Figure 7c. Since the laser energy and bubble energy were depleted during the laser-induced breakdown of the liquid and the first evolution cycle, the impact of the collapse shock wave generated during the second and third evolution periods on the specimen surface was relatively minor.

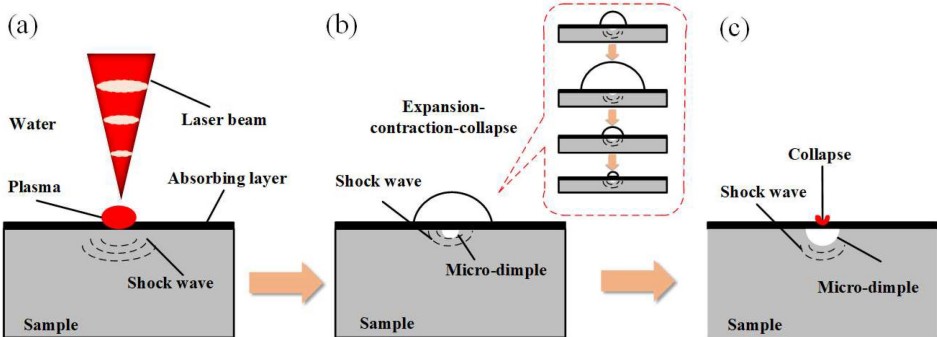

**Figure 7.** Principle of LICB impact on micro-dimple formation. (**a**) breakdown (**b**) expansion-contraction-collapse (**c**) final collapse.

*3.3. The Influence of Process Parameters on the Surface Morphology of Micro-Dimple*

3.3.1. The Influence of Absorption Layer on Surface Morphology of Micro-Dimple

After three repeated LICB impacts on different absorption layers, the three-dimensional morphology of micro-dimples on the sample surface is shown in Figure 8. Among them, specimens 1 to 3 were used without an absorption layer and black paint and copper foil as absorption layers, respectively. It can be observed from Figure 8 that the surface micro-dimple on specimens 1 to 3 all exhibit an approximately spherical crown shape, further demonstrating that the shock wave induced by laser cavitation bubble has a Gaussian distribution in the spatial scale. The repetitive loading of impacts results in the edges of the impacts being subjected to shear forces, leading to restricted plastic flow of the sample and volume transfer on the specimen surface. It can be observed from Figure 8a that due to the absence of an absorption layer, specimen 1 exhibits a typical "volcano-like" protrusion around the dimple, similar to the phenomenon observed in laser shock micro-texturing experiments conducted by the American scholar Caslaru [36]. Black paint, as an absorption layer, possesses strong laser absorption; however, its brittleness leads to cracking and ablation after repeated impacts. It can be observed from Figure 8b that the dimple on the surface of specimen 2 appears to be approximately axisymmetric, with a relatively smooth edge and few minor protrusions. Copper foil, serving as the absorption layer, mitigates laser-induced thermal effects on the specimen surface during the formation of the micro-dimple by LICB. It can be observed from Figure 8c that the circular micro-dimple on specimen 3 is axisymmetric, with a smooth and flat edge devoid of protrusions.

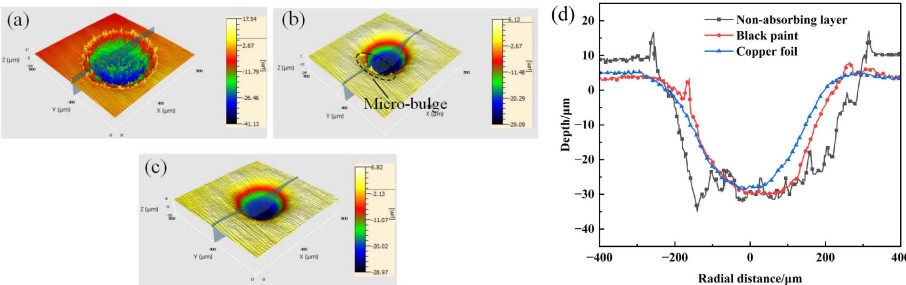

**Figure 8.** Surface morphologies of micro-dimple under different absorption layers. (**a**) Non-absorption layer; (**b**) black paint; (**c**) copper foil; (**d**) two-dimensional cross-section diagram.

To compare the micro-dimple's morphology induced by laser cavitation bubble with different absorption layers, the cross-sectional profiles of the micro-dimples were captured. The 2D sectional comparison of the micro-dimples for specimens 1 to 3 are shown in Figure 8d. It can be observed from Figure 8d that the "volcano-like" protrusion in the cross-section of the micro-dimple of specimen 1 is the most prominent, with a rough bottom. Micro-dimple in the cross-section of specimen 2 exhibits slight protrusions, with a smooth bottom that is approximately axisymmetric. Micro-dimple in the cross-section

of specimen 3 shows no protrusion, with a smooth and flat bottom presenting geometric circular symmetry. The depth of the micro-dimples on the cross-sections of specimens 1, 2, and 3 are −34.91 µm, −29.99 µm, and −28.47 µm, respectively. The depth of the micro-dimple is closely related to the laser absorption rate of the absorption layer. Black paint exhibits a higher laser absorption rate compared to copper foil. Under the same laser e energy and number of impacts, the diameters of the micro-dimples on the cross-sections of specimens 2 and 3 are approximately equal, with the depth of the micro-dimple in specimen 2 slightly deeper than that of specimen 3. Due to the absence of an absorption layer, specimen 1 surface experienced ablation, resulting in larger diameters and depth of the micro-dimple compared to specimens 2 and 3. In conclusion, the use of copper foil as the absorption layer in specimen 3 yields the best micro-dimple effect.

### 3.3.2. The Effect of Laser Parameters on the Surface Morphology of Micro-Dimple

Based on the experimental results of Section 3.3.1, a 100 µm thick copper foil was selected as the absorption layer. The influence of laser parameters on the diameter and depth of the micro-dimple was studied by varying the laser energy and the number of impacts. The contour profiles of the micro-dimple induced by LICB impacts under different laser parameters are shown in Figure 9. The cross-sectional morphology of micro-dimples obtained by repetitively subjecting them to impacts with different laser energy levels three times is shown in Figure 9a. It can be observed from Figure 9a that under the influence of LICB at varying laser energy levels, the maximum diameter of the upper part of the micro-dimple is approximately the same, at around 450 µm. As the energy distribution of the laser spot follows a Gaussian distribution, the laser plasma shock wave and cavitation collapse shock wave were unevenly distributed. The pressure of the shock wave decreased as one moves closer to the edge of the laser spot, resulting in the micro-dimple diameter being slightly lower than the spot diameter of 500 µm. Among these, when the energy ranges from 100 to 400 mJ, the depth of the micro-dimple gradually increases. However, when the laser energy reaches 500 mJ, the maximum depth of the micro-dimple begins to decrease and deviate from the center of the beam spot. Based on the studies by scholars Sollier [37] and Wang [38], it can be inferred that the plasma shielding effect impedes the effectiveness of laser radiation. In conclusion, the optimal effect of the LICB micro-dimple is achieved with a laser energy of 400 mJ.

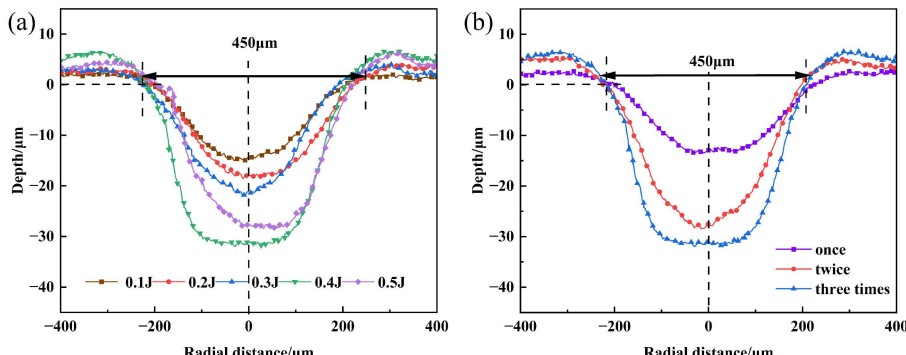

**Figure 9.** The micro-dimple plastic deformation depth curve with different laser energies and number of impacts. (**a**) Different laser energy; (**b**) different number of impacts.

Using a laser with an energy of 400 mJ, the cross-sectional morphology of micro-dimples obtained by varying the number of impacts is shown in Figure 9b. It can be observed from Figure 9b that under the impact of laser-induced cavitation with different numbers of impacts, the central plastic deforming depth of micro-dimples on the sample surface are −13.52 µm, −28.46 µm, and −31.99 µm, respectively. The amplitude increases in micro-dimples on the sample surface are 110.5% and 12.41%, with the rate of increase slowing down. The bottom of the dimple is sharper after 1–2 impacts and becomes smoother

after three impacts. According to the study conducted by Gu et al. [22], it is speculated that the observed phenomenon is as follows: when the laser energy is 100–300 mJ, even after three repeated impacts, the material's limit for plastic deformation is not reached. Moreover, due to the Gaussian distribution of laser energy on a spatial scale, the bottom of the micro-dimple formed is relatively sharp. Conversely, when the laser energy reaches 400 mJ, the central region subjected to repeated loading reaches the plastic deformation limit first, while the shear forces exerted on the surrounding area of the central region decrease, resulting in reduced compression with the sample surrounding the bottom of the dimple. Ultimately, the entire bottom of the dimple becoming smoother. Under this process parameter, the internal quality of the dimple was improved, and its volume increased, providing sufficient space for oil lubrication in micro-texturing.

In summary, the optimal effect was achieved by impacting three times with an energy of 400 mJ. The micro-dimple obtained after LICB maintained good circularity and geometric symmetry, indicating that consistent geometric features of a micro-dimple can be obtained by quantitatively controlling LICB technology parameters. The micro-dimple is repeatable, providing a theoretical basis and technological guidance for subsequent research on surface texturing of 7050 aluminum alloy induced by laser-induced cavitation.

## 4. Conclusions

The mechanism of near-wall laser-induced bubble micro-dimple formation on 7050 aluminum alloy is investigated. The effects of different process parameters of the laser-induced bubble on the surface morphology and depth of micro-dimple are analyzed. The following conclusions are drawn:

(1) By laser-induced cavitation bubble (LICB) technology, under the synergy of laser plasma shock wave and bubble collapse shock wave, a micro-dimple with a diameter of 450 μm and high surface quality is formed. The feasibility of the apparatus is verified, and a model of laser-induced bubble micro-dimple formation is established, elucidating the mechanism of micro-dimple formation fabricated by LICB.

(2) Copper foil, as an absorption layer, results in smooth, unburned, and protrusion-free micro-dimple induced by LICB. With laser energy ranging from 100 to 500 mJ, the depth of micro-dimple exhibits a trend of initially increasing and then decreasing, ranging between 15 and 35 μm. Under the action of a laser-induced bubble for 1 to 3 times, the plastic deformation depth of the micro-dimple shows a non-linear increasing trend between 10 and 35 μm, with the increasing trend slowing down, demonstrating saturation effects. The optimal process parameters for preparing a micro-dimple by laser-induced bubble are as follows: copper foil is the absorption layer, energy is 400 mJ, and the single point impact is three times.

(3) Due to limitations in the experimental equipment, the current high-speed camera fails to clearly capture the shock wave and water jet induced by LICB. In the future, more detailed studies on the mechanical effects of laser-induced shock wave and water jet can be conducted by new devices. Subsequent work also involves detecting residual stress and hardness on the surface of the micro-dimple to further explain the influence of the number of laser cavitation events on the smoothness of the dimple bottoms. In the future, should thicker copper foils become available, a greater number of experiments with repeated impacts will be conducted. In conclusion, LICB provides a new technical way for laser micro-texturing, with greater flexibility and precision, and provides a broad prospect for intelligent equipment friction reduction.

**Author Contributions:** Y.C.: Conceptualization, Methodology, Writing—review & editing. R.H.: Writing—original draft, Investigation, Data management, Writing—review & editing. W.S.: Project support, Supervision. R.Z.: Methodology, Writing—review & editing. All authors have read and agreed to the published version of the manuscript.

**Funding:** This work was supported by the Postgraduate Research & Practice Innovation Program of Jiangsu Province (KYCX22-3337), the Foundation of National Key Laboratory for Remanufacturing

(1420052022KJW01), the Project of Laser Processing and Metal Additive Manufacturing Technology and Application (SJK2023-3), Nantong Municipal People's Livelihood Project (MS22022040).

**Informed Consent Statement:** Not applicable.

**Data Availability Statement:** No new data were created or analyzed in this study. Data sharing is not applicable to this article.

**Conflicts of Interest:** The authors declare that they have no known competing financial interests or personal relationships that could have appeared to influence the work reported in this paper.

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
