# Peer review of "Experimental Study on Near-Wall Laser-Induced Cavitation Bubble Micro-Dimple Formation on 7050 Aluminum Alloy"

_water, doi:10.3390/w16101410_

Round 1

Reviewer 1 Report

Comments and Suggestions for Authors

In this paper, the feasibility and forming law of laser-induced cavitation bubble fabricating micro-dimple on the surface of 7050 aluminum alloy were studied. The study achieved quantitative loading and precise control of the micro-dimple forming process, providing a new technological method for laser micro-texturing. However, there are also some problems in this paper that need to be corrected by the author, stated as follows:

1. The laser-induced cavitation bubble technology is developed from underwater laser ablation technology. Please provide further details on its correlation with and distinctions from laser ablation technology.

2. Can the experimental results be replicated? As it is well known, numerous factors influence the growth and collapse of cavitation bubble. Moreover, are there any additional factors contributing to variations in the maximum diameter and evolution time across multiple periods?

3.The time step between two images in Figure 3 should be a multiple of the delay between two consecutive image acquisitions (25 μs for 40000 fps). However, the time step appears inconsistent in the figure 3. Does this imply that the presented image sequence is composed of multiple experiments with different acquisition speeds? This should be explicitly stated.  

4. Through three repeated laser impacts with different energy, the plastic deformation depth curve of the micro-dimples shown in Figure 10(a) was obtained. Why is it that only under the influence of 400mJ laser energy, the bottom of the micro-dimple on the sample tends to become smoother? Please provide a detailed explanation.

5. The distinction in Figure 9Figure 10 in the paper are not obvious enough, and it is recommended to redraw the figure to improve the recognition of the curves in the figure.

6.The references must be updated, with more published relevant articles should be cited.

Comments on the Quality of English Language

The wording and style of some section need careful editing.

Reviewer 2 Report

Comments and Suggestions for Authors

This manuscript has done meaningful work on investigating the feasibility and forming laws of fabricating micro-dimple induced by near-wall laser-induced cavitation bubble on 7050 aluminum alloy. The structure is complete and the results of this study do provide some guidance for the subsequent micro-texture manufacturing with this process. This manuscript should undergo a major revision and it will be accepted for publication after considering the following recommendations.

1.      It is necessary to rewrite the abstract. There are some repetitive statements.

2.      In introduction, it is necessary to extend the application of collapse of cavitation bubbles for material removal process, such as IJMS 257 (2023) 1085.

3.      Since the process mentioned in this paper is for processing microtextures, why not introduce other microtexture preparation processes in the introduction section?

4.      The working position and holding mode of the hydrophone in the device site diagram need to be shown.

5.      How to explain the negative pressure part of the signal received by hydrophone?

6.      Figures 8 and 9 would be better together in one figure.

7.      When studying the laser energy you chose 5 gradients to get the best laser energy (400mJ), which makes sense. But when studying the number of processing times, why did it stop after only three times? It is clear that 3 times has not reached the processing limit.

8.      The font size in the figure should be smaller than the font size of the text content. Make changes to some diagrams (such as Figure 7).

9.      The format of the references is not consistent. Please make the necessary changes according to the requirements of the journal.

10.   Please check the whole manuscript to ensure that there are no grammatical or typographical errors.

Comments on the Quality of English Language

Good!

Reviewer 3 Report

Comments and Suggestions for Authors

The manuscript entitled Experimental study on near-wall laser-induced cavitation bubble micro-dimple forming of 7050 aluminum alloy is the experimental study of the laser-induced cavitation. I commented as follows;

1.Introduction is not clear. The author should revise it. Especially, the disadaventages of the previous studies and advantages of the present study should be shown.

2.As shown in Fig. 5, relatively small bubble was generated. If I increase the time, will the particle size get smaller?

3.The many symbols and letters are used. The author should summarize them as a nomenclature.

Comments on the Quality of English Language

No comment.

Round 2

Reviewer 1 Report

Comments and Suggestions for Authors

In the conclusion section, the author can add some information about the applicability  and uniquenessof the LICB technolgy.

Comments on the Quality of English Language

In the conclusion section, the author can add some information about the applicability  and uniquenessof the LICB technolgy.